# Balance Rehabilitation Approach by Bobath and Vojta Methods in Cerebral Palsy: A Pilot Study

**DOI:** 10.3390/children9101481

**Published:** 2022-09-28

**Authors:** Andreea Ungureanu, Ligia Rusu, Mihai Robert Rusu, Mihnea Ion Marin

**Affiliations:** 1Faculty of Physical Education and Sport, University of Craiova, 200585 Craiova, Romania; 2Faculty of Mechanics, University of Craiova, 200585 Craiova, Romania

**Keywords:** cerebral palsy, balance, therapy activation, rehabilitation

## Abstract

In cerebral palsy (CP) the basis for rehabilitation comes from neuroplasticity. One of the leading therapeutic approaches used in the management of CP is the NDT Bobath therapy and Vojta therapy consists in trying to program the ideal movement patterns for the age. The aim of our research was to analyze, from a functional point of view, the evolution of the biomechanical parameters characterizing the balance, in children with CP. The group of 12 subjects average age of 7 ± 3.28 years. The subject’s evaluation included a functional clinical evaluation by Berg pediatric scale and a biomechanical evaluation performed using the “Stabilometry footboard PoData 2.00” for evaluation the body weight distribution on the foot level. The rehabilitation program was developed based on two methods, NDT Bobath and Vojta. A 90-min physiotherapy session starts with a Vojta therapy activation, for 20 min. Between the two therapies there is a 10-min break, then the session continues with NDT Bobath exercises within the 3 physical exercises proposed for 60 min. 5 days per week, 6 months. The analysis of the data collected before and after the application of the rehabilitation program, regarding the using the Berg scale indicates a progress of 32.35%, (*p* = 0.0001 < 0.05) and the effect size is large. The evolution of the data that indicate the distribution of body weight at the level of the two lower limbs, at the two moments pre/post, evaluation. For left side a progress of 8.39%, (*p* = 0.027 < 0.05) but a small effect size of 0.86. For right side a progress of 10.36% (*p* = 0.027 < 0.05) and also a small effect size of 0.86. Analyzing the results, we find that there is a left-right rebalancing in most patients. The favorable results that were obtained by drawing up a physiotherapy program composed of the combination of the two Vojta and NDT Bobath methods are proof of the fact that both methods are based on the creation of a stimulating peripheral pressure, which, if maintained, generates an extended stereotyped motor response. A pattern of symmetrical muscle contraction is thus created and thus balance and postural control can be achieved. The left-right rebalancing, proven by the percentage distribution analysis of the weight at the lower segmental level, demonstrated that the body alignment approach through the Vojta method on the one hand and the inhibitory facilitating postures/exercises promoted by the NDT Bobath method, allows obtaining a symmetry.

## 1. Introduction

Neuroplasticity refers to the central nervous system’s ability to change, and this change is not singular for central nervous system (CNS) injuries or in response to rehabilitation protocols. Neurons have the ability to change their structure and function depending on the inputs generated by activity and learning; in fact, neural change is the basis for memory and behavioral change that results from experience [1]. Plasticity takes place constantly, regardless of whether we are subjected to intense training or not. In addition, plasticity can be positive (adaptive) or negative (maladaptive). The central nervous system (CNS) has an innovative capacity to recover and to adapt to the compensatory mechanisms following an injury [2]. The basis for rehabilitation comes from neuroplasticity, defined as the ability of neural network to make adaptive changes at both structural and functional levels, ranging from molecular, synaptic and cellular changes to global network changes [3]. Human balance and gait are the most complex tasks that require the coordination of different neural network in the brain, brainstem, cerebellum and spinal cord [4] and they can recover function after injury/insult to any of these structures by applying methods that indirectly use neuroplasticity. Moreover, the activity of muscles and joints involved in walking and balance will be controlled via the descending motor tracts [5], while and important intervention for gaining balance is achieved through the pyramidal tracts (corticospinal tract) and the extrapyramidal tracts (vestibulospinal tract) [6,7].

Cerebral palsy (CP) is a pathology that may involve an interruption of the descending tracts. It is caused by a non-progressive disorder of the brain that occurred during the fetal or developing infant period. This disorder can cause certain permanent limitations of posture and movement [8]. One of the major challenges for children with infantile cerebral palsy is spasticity which causes a deficit of balance and gait [9]. In these cases, neuroplasticity is the key to functional recovery. Physiotherapists therefore need to understand the impact of rehabilitation on neural reorganization during recovery. Many of the neurosciences researched over the past 3 decades and more recently have been focused on understanding the physiological mechanisms underlying neuromotor recovery, as well as its determinants, so that we can maximize the effects of applied treatments [10]. However, there are fewer studies that have reported the results of objectifying the application of recovery methods that activate neuroplasticity.

There is no unique way to action for curing infantile cerebral palsy (CP) or for eliminating brain lesions [11], but there are therapeutic methods such as NDT Bobath therapy [12] and Vojta therapy [13] that, through reflex mechanisms of neural stimulation, as a result of reflex postures or specific mobilizations, contribute to improving balance.

Neuroplasticity is stimulated by intense repetitive practice, but it should not exhaust the nervous system. It is necessary for the treatment not to focus exclusively on repetition, but to take into account the patient’s mobility and activity levels. Studies of the brain response of a brain-injured child have provided important information and diverse opinions concerning the effects that age has on recovery. The Kennard Principle, first advanced by Margaret Kennard, has demonstrated the fact that the developing brain is capable of significant reorganization and recovery after trauma [14]. In addition, the younger brain, in contrast with the older one, is less likely to develop progressive cognitive decline [15].

The greatest potential to obtain a good evolution in children with CP, based on neuroplasticity, is found in children whose rehabilitation intervention started early. This is because the gross motor skills in children with CP stabilize at the age of 4–7 years, while after this period the motor skills are relatively stable. Up to this age, we are dealing with a critical period in which the effects of neuroplasticity are exerted in relation to the motor control centers in the brain [15].

Analyzing the scientific literature, in order to document the main therapeutic interventions in children with CP, shows that one of the leading therapeutic approaches used in the management of brain neuroplasticity underdevelopment is the NDT Bobath therapy [12] (Neurodevelopmental Therapy), which is based on the fact that the atypical development concerning postural control and reflexes is responsible for the observed motor anomalies, due to the basic dysfunction of the central nervous system. This approach aims to facilitate standard motor development and functioning and to prevent the development of secondary disorders caused by muscle contractures and joint deformities of the limbs. The basic principle is that any stimulus of supraliminal intensity can “awaken” a clinical sign of CNS [16]. All therapeutic interventions based on the concept of neurodevelopment were built with the aim of improving movements and the active participation in maintaining body alignment and stabilizing each part of the body. The exercise program proposed by most authors was designed to lengthen the extensor muscles of the lower limbs with a greater focus on the hip muscles [17].

Despite the widespread use of NDT Bobath therapy there is no rigorous research regarding its clinical effectiveness. NDT aims at the normalization of muscle tone, its primitive and abnormal inhibition and the facilitation of normal movements. The goal of this approach is to correct postural tone abnormalities and facilitate more regular movement patterns for performing daily activities. Despite its widespread use in the clinic, limited studies have demonstrated evidence for its effectiveness in children with CP. Thus, a valid and reliable assessment tools are needed to measure the effectiveness of therapy in CP rehabilitation [18].

Following the four main patterns present at the level of the lower limbs of children with infantile CP, spastic form, a method was developed aiming to overcome spasticity with the help of spinal reflexes, by using “reflex unlocking” techniques, so that through peripheral stimulation the higher nerve centers are activated, as their activity is modeled on the basis of neuroplasticity. The techniques were adopted by Doman, and later by V. Vojta, whose approach was based on combining the techniques of Kabat and Fay [12]. According to Vojta, the role of therapy consists in trying to program the ideal movement patterns for the age of the newborn and of the infant with an affected central nervous system, as much as it is possible. This means that neurophysiological programming attempts to introduce an automatic coordination of the body’s position, with well-defined angles of the upper and lower limbs relative to the trunk and vice versa, and of different body parts to each other, in a regular and reciprocal manner (alternatively on both sides of the body, left and right), with a change in the center of gravity’s position, as it is common with each movement [19].

At a neurophysiological level, stimulation via Vojta therapy [13] induces psychomotor changes [20]. Vojta therapy [13] is a method that suppresses abnormal movement and induces normal motor development by promoting postural control and suppressing the patient’s compensatory movements that were wrongfully acquired [21].

Vykuntaraju Kammasandra Nanjundagowda (2014) notes that medical recovery in the pathology of cerebral palsy aims to promote normal movement patterns and inhibit abnormal ones to maximize the independence of motor functionality. He mentions that several physical therapy techniques can be used in the rehabilitation plan: “Bobath Neurodevelopmental technique, the Vojta method, the Kabat, Philps and Denver method, but there is no clear evidence to support the superiority of one over the other” [22].

In the recovery process of children with infantile CP, there have been several major therapeutic practices in recent years, including the NDT Bobath concept and Vojta therapy; these treatment models have been adopted as best practices and accepted as conventional treatment approaches. However, additional, well-regulated, randomized studies are needed in order to determine the efficiency and the most appropriate roles for new technologies in physical rehabilitation interventions [13].

The aim of our research study was to analyze, from a functional point of view, the evolution of the biomechanical parameters characterizing postural control and balance, in children suffering from CP, aged between 3 and 11 years old and who were included in physical therapy programs consisting of exercises based on the principles and means of the NDT Bobath and Vojta methods. We chose this approach because currently there is no research to provide actual results regarding the quantification of the combined application of the two methods in the recovery of children with CP.

## 2. Materials and Methods

### 2.1. Subjects

We studied a number of 12 subjects diagnosed with CP, tetrapharesis, aged between 3–11 years. The small number of subjects is due to the fact that the consistent participation of these children in a rehabilitation program is limited. Therefore, we cannot cover a complete application of rehabilitation program. The group of subjects included 4 boys and 8 girls, with an average age of 7 ± 3.28 years, average height of 121.7 ± 22.48 cm, average weight of 28.68 ± 15.95 Kg (mean value ± SD). Score I GMFCS for 3 children, score II for 6 children, score III for 3 children. The research has been approved by Etich Committee of Research Center-Human Body Movement Research Center (approved number 1540/1.10.2021) and respect the rules of Helsinki Declaration about the research that included human subjects. In the same time we have the informed consent for all children that have been included in the research. The informed consent has been signed by their parents.

The values of the anthropometric measurements for the group of subjects are those recorded in the table below (Table 1):

The selection of the 12 subjects was made based on inclusion and exclusion criteria as follows:

*Inclusion criteria*:children diagnosed with CP;children aged between 3–11 years;children capable of understanding and executing commands;children without other associated diseases;children who did not participate in recovery programs that were based on the two methods, NDT Bobath and Vojta;children with GMFCS level I-IV;children who can adopt the orthostatic position necessary to assess balance.

*Exclusion criteria*:children with mental retardation;children who cannot participate constantly in physical therapy sessions;children with deformities of the locomotor system;children with visual or hearing impairments;children with spasticity more then 2 on Ashworth scale.

### 2.2. Evaluation of Subjects with CP

The subject’s evaluation included a functional clinical evaluation and a biomechanical evaluation. The clinical functional evaluation of the 12 subjects included the balance assessment using the Berg scale [23]. The biomechanical evaluation was performed using the “Stabilometry footboard PoData 2.00” bipodalic platform with a built-in podoscope that can be directly connected to a computer via USB ports. The device is represented by six load cells that can be positioned to detect the distribution of body weight at the points corresponding to the 1st metatarsal, the 5th metatarsal and the heel of each foot. It is also used to measure the average position of the body’s center of gravity and its small movements around this position. This equipment works based on the principle of gathering information from the plantar level, and the data provided comes from the stimulation of the platform’s sensors [24].

The patients were prepared before the start of the assessment, being minimally dressed, as the assessment was carried out in basal conditions. At the time of evaluation, we must take into accounted the fact that the CNS system, through its extero-proprioceptive receptors, is able to identify the best postural strategies, for each moment, adapting them to the contingent situation. Regarding the vertical position, its efficiency is determined by the distribution of body weight on both legs. For this reason, three measurements were made.

The patient must be without shoes, the skin must be bare. Alone he/she steps on the platform. The platform is divided into two equal parts, the child puts one foot on the right side and one foot on the left side, and must remain still for 20 s (Figure 1). The result is the footprint obtained as shown in Figure 2.

This platform supported the analysis of some parameters such as the left-right weight distribution, a feature that we believe is relevant for assessing the evolution of balance depending on the regional loading at the plantar level. The subjects’ evaluation was carried out at two times—evaluation 1 (EV1) and evaluation 2 (EV2), 6 months apart.

### 2.3. Statistical Analysis

For statistical analysis we used descriptive analysis and regarding normal distribution of the parameters we applied JB test (Jarque–Bera). Student’s t-test was used for analysis the differences between parameters values and give information about how was the evolution of the parameters from first moment of evaluation (EV1) and second moment of evaluation (EV2) Student’s *t*-test was applied for equal means and shows if is a significant difference. The effect size was assessed by Cohen D coefficient [25,26]. We used XLSTAT software for statistical analysis.

### 2.4. Rehabilitation Program

The rehabilitation program was developed within a general framework that took into account certain requirements imposed by the application of the two methods, NDT Bobath and Vojta. The patient’s positioning is the element from which the therapeutic exercise is initiated. For the NDT Bobath technique/method, exercises, were performed in supine position (SP), lateral position (LP), prone position (PP), on all fours, on both knees, sitting on the edge of the bed or sitting on the chair, the half kneeling position, and orthostatism position. For the Vojta method, exercises that require activation positions in SP, LP, PP, and the first position (crouching at the edge of the bed) were performed.

The program’s structure includes 10 NDT Bobath exercises and 3 Vojta exercises (activations). The program started with 5 daily sessions, for a period of 3 weeks, followed by 3 sessions per week performed by a physiotherapist in the office and 2 sessions performed by the parent at home.

Each session is scheduled for 90 min, the program being carried out for a period of 6 months.

Methodology: A 90-min physiotherapy session started with a Vojta therapy activation, for 20 min, between the two therapies there is a 10-min break, then the session continued with NDT Bobath exercises for 60 min each day, 5 days per week, for 6 months. The physical therapy program was carried out in the following stages: Vojta therapy, on the physical therapy table, activated from three positions: SP, LP, PP, activation time—5 min in each position, a total of 20 min of Vojta activation; NDT Bobath therapy, which took place in the following stages: on the physiotherapy table: SP, LP, PP; on the mattress: on all fours, on both knees, in orthostatism; on wall bars, on the NDTBobath ball; on the walking treadmill. The therapy is carried out on an outpatient basis, with the patient coming to the office every day to benefit from the therapy.

On first session, patient assessment was performed using scale application and PoData platform. Following the evaluation, we established the functional diagnosis, the degree of segmental and global mobility, balance and the ability to move.

The first session also aimed to initiate the exercise program, information will be provided concerning the objectives of the treatment plan and how it will proceed, the timetable of the upcoming sessions is also set according to the therapist’s and the patient’s schedule. The physiotherapist will provide information about the purpose, the objectives of the therapies addressed in the recovery plan, NDT Bobath and Vojta, what the therapies entail, the importance of performing exercises at home and the required work items. The parent or caregiver has been instructed on the execution of certain elements of the Vojta therapy and the kinesiological responses expected. The child and the caregiver received homework, light exercises, essential for achieving the goals, in order to maintain the gains made in the physical therapy sessions. Recommendations were given to correctly position the child during daily activity. The duration of the session is 50–60 min.

#### 2.4.1. Vojta Therapy

Vojta therapy had as its therapeutic objective the improvement of the body’s balance during movements (“postural coordination”), an objective for which I built and applied the following two physical exercises:


**Exercise 1. Reflex crawling**


*The objective of the exercise*: It is a movement complex that is made up of the essential components of movement: posture coordination; lifting against gravity, targeted and precise movements of the arms and legs. Thus, reflex crawling possesses the basic patterns of locomotion.

*The execution method*: the child is positioned in the ventral decubitus, the head is placed on the work surface and turned to one side, with the occipital part towards the physiotherapist, being called the occipital part, and the facial part looks forward.

The movement takes place into the so-called cross scheme, in which the right leg and the left arm move at the same time, as well as vice versa. One leg and arm on the opposite side supports the body and moves the trunk forward.

In therapy, when the child begins to rotate the head, the therapist applies appropriate resistance. This increases the activation of the entire body’s musculature, thereby achieving the prerequisites for the verticalization process.

*The execution technique*: will activate from this position 3 times on each side, for a 15–50 s activation. We start with the activation on the right facial side, the child is in the prone position, with the right upper limb (UL) in 180° flexion at the shoulder level and a slight flexion at the elbow level. Lower limb (LL) on the occipital side (left LL) flexed to 45° and flexed at the knee. Left upper limb and right lower limb are in extension. From this position we press on the “calcaneus” area at the LL level on the occipital side and the “epicondyle” area at the elbow level on the facial UL pressing gently, it remains in position pressing continuously and slowly for 15–50 s, until the expected kinesiological response is obtained: at the level of UL on the facial side to obtain an extension of the fingers, LL on the facial side performs the flexion towards the chest, LL on the occipital side performs flexion at the level of the fingers, UL on the occipital side initiates the flexion movement. After the kinesiological response is obtained, the activation side is changed, now the facial side will be the child’s left side. These activations are repeated 3 times for each part (Figure 3).


**Exercise 2. First position**


*The objective of the exercise*: Lifting against gravity, extension of the spine, targeted and precise movements of the arms and legs.

*The execution method*: the child is positioned in a crouching position, on the edge of the bed, with the seat on the heel, and the legs are positioned outside the physiotherapy table in order to monitor the kinesiological response and not to be restrained—position represented in Figure 4. The head is placed on the surface work and turned to one side, with the occipital part towards the physiotherapist, being called the occipital part, and the facial part looks forward. The facial upper limb is in 180° flexion, and the occipital upper limb is in extension.

The movement takes place in a so-called cross scheme, in which the right leg and the left arm move at the same time, as well as vice versa. One leg and arm on the opposite side supports the body and moves the trunk forward.

In therapy, when the child begins to rotate the head, the therapist applies appropriate resistance. This increases the activation of the entire body’s musculature, thereby achieving the prerequisites for the verticalization process.

*The execution technique*: will activate from this position 3 times on each side, for a 15–50 s activation. We start with the activation on the right facial side, the child is in the prone position, with the right upper limb (UL) in 180° flexion at the shoulder level and a slight flexion at the elbow level. From this position we press on the “calcaneus” area at the LL level on the occipital side and the “epicondyle” area at the elbow level on the facial UL pressing gently, it remains in position pressing continuously and slowly for 15–50 s, until the expected kinesiological response is obtained: at the level of UL on the facial side, an extension of the fingers is obtained, UL on the occipital side initiates the flexion movement, the leg on the facial side performs an eversion movement with extension and abduction of the fingers, the leg on the occipital side performs an inversion with flexion of the fingers. After the kinesiological response is obtained, the activation side is changed, now the facial side will be the child’s left side. These activations are repeated 3 times for each part.

*Dosage*: 2 activations per each part;

*Sets*: 3 sets, no rest between sets.

#### 2.4.2. NDT Bobath Therapy

NDT Bobath therapy had as its objective the re-education of balance, coordination and balance obtained within the 3 physical exercises proposed and applied.

**Exercise 1. Quadruped imbalances** (Figure 5)

*The execution method*: The subject is positioned on all fours, the therapist imbalances him by pushing him sideways and backwards from the shoulders and sideways and forwards from the pelvis. The amplitude and force that the physical therapist applies to imbalance the subject is dosed according to the subject’s ability to rebalance.

*Dosage*: 10 repetitions;

*Sets*: 2, with 30 s rest between sets.

**Exercise 2—Imbalances from the “kneeling” position** (Figure 6)

*The execution method*: With the subject supported on his knees, the therapist imbalances him from all directions. The force and amplitude of the imbalance is dosed according to the subject’s ability to rebalance.

*Dosage*: 5 repetitions;

*Sets*: 2, with a 30 s break between repetitions.

**Exercise 3—The Cervant Knight** (Figure 7)

*The execution method*: The subject loads on one knee, the contralateral lower limb performs a triple flexion (flexion at the hip level, flexion at the knee level, and flexion at the ankle joint level)—the position of the serving knight (pelvis in retroversion on the side without load), inhibits spasticity of the adductors and extensors of the hip on the non-loading side, and the flexors of the hip on the loaded side, facilitating stabilization of the pelvis.

*Dosage*: 5 repetitions per each leg;

*Sets*: 2, with a 1-min break between sets.

## 3. Results

### 3.1. Functional Clinical Evaluation

The analysis of the data collected before and after the application of the rehabilitation program, regarding the assessment of balance using the Berg scale for pediatrics, indicates an improvement in the Berg scale score, the results being reproduced in Figure 8 and Figure 9 for the 12 subjects.

The evolution from a clinical functional point of view is in the sense of an improvement in the balance evaluated by the Berg scale, more accentuated in patient P3, P5 and P10.

The results of the statistical analysis of the clinical functional evaluation by means of the Berg scale are presented in Table 2, Table 3 and Table 4.

The statistical analysis indicates a progress of 32.35%, with an average increase in the Berg scale value of 9.16. Two-sided *t*-test reveals a statistically significant difference in averages *p* = 0.0001 < 0.05, for t_obs_ = 7.99 and df = 11. It is observed that the effect size is large.

Regarding evolution of GMFCS score we present the results in the next figure (Figure 10 for P1–P6 and Figure 11 for P7–P12).

### 3.2. The Results of the Biomechanical Evaluation

In this paper, we present the evolution of the data that indicate the distribution of body weight at the level of the two lower limbs, at the two moments of the EV1 and EV2 evaluation.

In order to evaluate the evolution as objectively as possible from the recovery of balance and implicitly the efficiency of the combined therapy point of view, we consider that it is necessary to focus our attention on how the patients manage to balance left and right.

In this sense, Table 5 shows the body weight distributions (expressed in kg and %) at the level of each leg, at the two moments of the evaluation (EV1 and EV2).

Analyzing Table 5, it is observed that all measured values have a normal distribution according to the Jarque-Bera test (*p* ≥ 0.05, so H0 is confirmed).

In order to carry out a deeper analysis, we considered it useful to follow how the weight is distributed at the level of each leg, expressed as a percentage of the body weight.

#### 3.2.1. Weight Distribution on the Left Leg

The distribution of the weight on the left leg shown in percentage in Figure 12, under the aspect of evolution between the two moments of the evaluation.

An evolution of the load on the left leg is observed, towards 50% of the body weight.

Table 6, Table 7 and Table 8 show the statistical indicators that highlight the degree of percentage distribution at the level of the left leg, percentage of weight.

The statistical analysis indicates a progress of 8.39%, with an average decrease of 4.64. The bilateral t-test reveals a statistically significant difference in averages, *p* = 0.027 < 0.05, for t_obs_ 2.552 and df = 11. A small effect size of 0.86 is observed.

#### 3.2.2. Weight Distribution on the Right Leg

The same aspect of the percentage distribution of body weight at the level of the right leg can be found in Figure 13.

The same evolution is observed in the sense of decreasing or increasing the distribution so that one can talk about an attempt to rebalance.

Table 9, Table 10 and Table 11 highlight the statistical indicators.

The statistical analysis indicates a progress of 10.36%, with an average decrease of 4.64. The bilateral t-test reveals a statistically significant difference in means, *p* = 0.027 < 0.05, for t_obs_ = 2.552 and df = 11. A small effect size of 0.86 is observed.

Analyzing the previous results, we find that there is a left-right rebalancing in most patients.

## 4. Discussion

The favorable results that were obtained by drawing up a physiotherapy program composed of the combination of the two Vojta and NDTBobath methods are proof of the fact that both methods are based on the creation of a stimulating peripheral pressure, which, if maintained, generates an extended stereotyped motor response. A pattern of symmetrical muscle contraction is thus created, at the level of the neck, trunk and limbs, as a result of the summation, and thus postural control can be achieved.

We found the clinical and functional expression of this aspect in the assessment of balance with the help of the Berg scale, which proves to be particularly useful in establishing the rehabilitation program, being an objective means of monitoring the evolution. This aspect of the importance of the Berg scale in the preparation of the balance rehabilitation program was also highlighted by Louie and colab [27] who used this scale in the admission stage of the stroke patient in the recovery program, considering that it is a predictive element of the evolution of balance and gait, which registered an important improvement 6–7 weeks after the initiation of the rehabilitation program. Thus, the authors consider that a patient who has a Berg score of at least 29 has a chance to regain his balance in 6–7 weeks of therapy. In the case of our study, we note that all subjects registered scores that exceed the value of 29 of the Berg scale score. Consequently, we can consider that the evolution of the subjects, objectified by this scale, confirms the fact that the complex therapeutic approach is effective and can be supported by concrete data.

The left-right rebalancing, proven by the percentage distribution analysis of the weight at the lower segmental level, demonstrated that the body alignment approach through the Vojta method on the one hand and the inhibitory facilitating postures/exercises promoted by the NDT Bobath method, allows obtaining a symmetry of the base and control of the trunk. It is known that there is a correlation between the asymmetry of the pelvis and the control of the trunk in children with PCI. The objectification of the effects of Vojta therapy was found in studies carried out in patients with stroke, in which it was observed that the Trunk Control Test (TCT) recorded an average value of 25 (Normal 0–43), compared to the result obtained in a group of patients who followed classical therapy and in which the average score was 46 [28].

We found in our research that there is a favorable evolution of weight distribution at the level of the two legs, with the achievement of a rebalancing with the increase of the load at the level of the right leg and the decrease at the level of the left leg, but what is more important is a closeness to the value of 50% within the rating from time EV2 as opposed to time EV1.

The effectiveness of the Vojta therapy was also reported by Hyungwon and Kim who noted positive results in the changes in each joint angle in the sagittal plane after the Vojta therapy. The conclusions of the study indicate that Vojta therapy can play an important role in improving the spatiotemporal parameters of the gait of children with spastic diplegia [22], this being a study that proves that through this approach an improvement in balance is obtained.

Proximo-distal stabilization in supine, prone, sitting and standing positions favors proprioceptive and vestibular stimulation.

Regarding the NDTBobath therapy, as I mentioned before, the objectification of the effectiveness of this therapy is minimal, the results of our study are supported by the study of Erdogan Kavlak (2018) who investigated the effects of this therapy after 8 weeks of application for the re-education of balance for children with CP [29]. The result was a significant improvement in balance, gross motor function and the level of functional independence.

NDTA study by Moazma Jamil compared effects of convention and NDT Bobath therapy to improve GMF among 24 children with CP. Children with CP received 3 months of intervention, 40 min/day, 5 days/week All children were tested with the Modified Ashworth Scale before starting treatment and 16 weeks after treatment. The result of the study showed that NDT Bobath therapy is more effective compared to conventional treatment [30]. The reduction of spasticity as well as the promotion of inhibitory reflex postures cause cortical activation, which, based on neuroplasticity, can trigger postural control and rehabilitation with recovery postural balance. This aspect was proven in our study by adjusting the left/right weight distribution, objectified by the percentage values of the loaNDT d.

The recovery process after brain injury is long, but with emerging evidence for neuroplasticity, the outlook for recovery is no longer so bleak. The exact mechanism remains unknown; however, many hypotheses are currently being investigated.

Zanon et.al. [31] in their paper consider that effect of NDT therapy is still uncertain and need more studies for assess the evolution of children with CP. Much more does not exist studies that reflects the results of NDT in balance rehabilitation.

Regarding Vojta therapy we found the paper of Mengibar et.al [32] that spoke about the role of Vojta therapy in accelerate the acquisition of GMFM-88-items and seems to activate the postural control. Furthermore, in this study this method seems to improve the balance in CP, but without biomechanical evaluation.

Our results demonstrate that there is a need to have a holistic approach of children with CP, means that also in clinical field the physiotherapist could apply combination of method for improve the movement and balance patterns. In the same time use of this evaluation, periodically could help the physician to monitoring the evolution and together with the physiotherapist design the rehabilitation goals.

We consider that these findings demonstrate the importance to use combination of the two techniques, but of course this combination could be applied to children with severed CP. For this reason, the physical therapist must have good knowledge about NDTBobath and Vojta methods. They have to be habilitated in field of these two methods and they will be able to design the interventions.

## 5. Conclusions

The training strategies promoted by the combination of the two methods are shown to be useful in the recovery program, which was reflected in the improvement of the Berg scale scores.

The results of the biomechanical evaluations complement the functional clinical evaluation and highlight as the main aspect the right recovery, which is in agreement with the Bobath method’s objectives, namely the fact that the efficiency of the method is to make the patient maintain his balance and distribute his weight bilaterally by reducing right/left asymmetry.

At the level of the foot, this selective control of the movement is very strong, a fact also demonstrated by the loading mode of the foot in the studied patients. We can talk about an element of prediction in terms of estimating the effectiveness of the therapeutic program.

The effect of the combined therapy, at the level of the foot that presents a behavior that seems to prepare the dynamic balance.

Left-right weight transfer must be carried out while maintaining postural control.

## 6. Limitations

The study has limitations due to the small number of research subjects. In addition, during the moment of measurements some of children have not full cooperation. The same could be say regarding the participation to the physical therapy program and how the children answer to the physical therapy tasks.

## Figures and Tables

**Figure 1 children-09-01481-f001:**
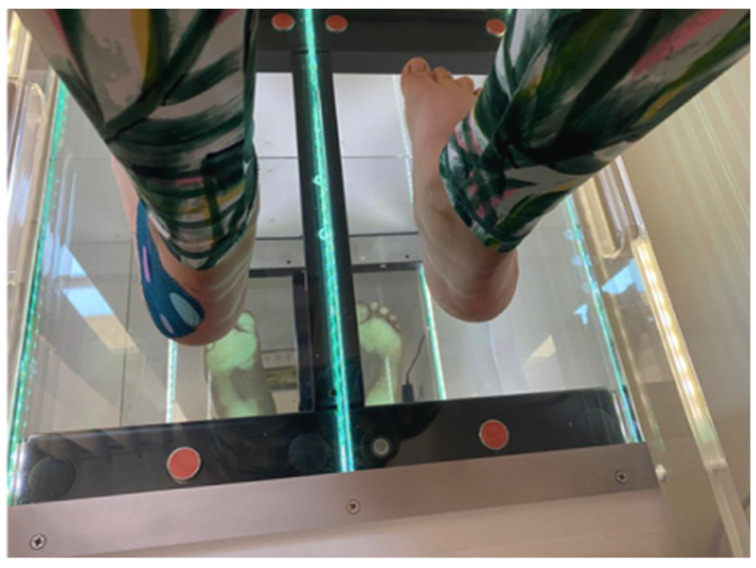
Patient preparation and positioning.

**Figure 2 children-09-01481-f002:**
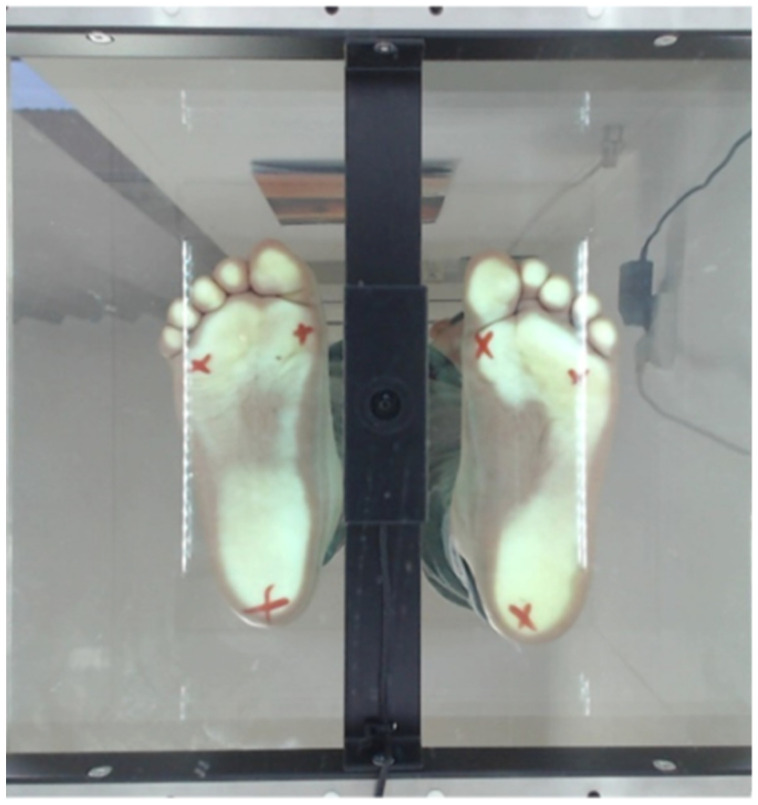
Plantogram picture (left spastic hemiparesis patient).

**Figure 3 children-09-01481-f003:**
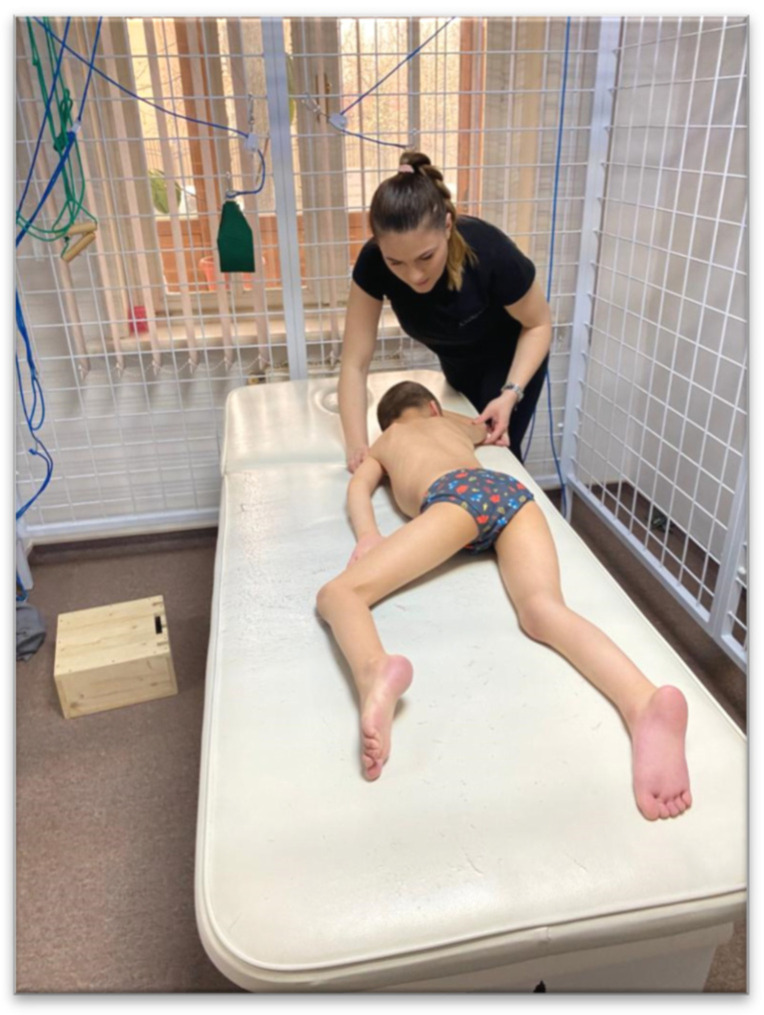
Reflex crawling.

**Figure 4 children-09-01481-f004:**
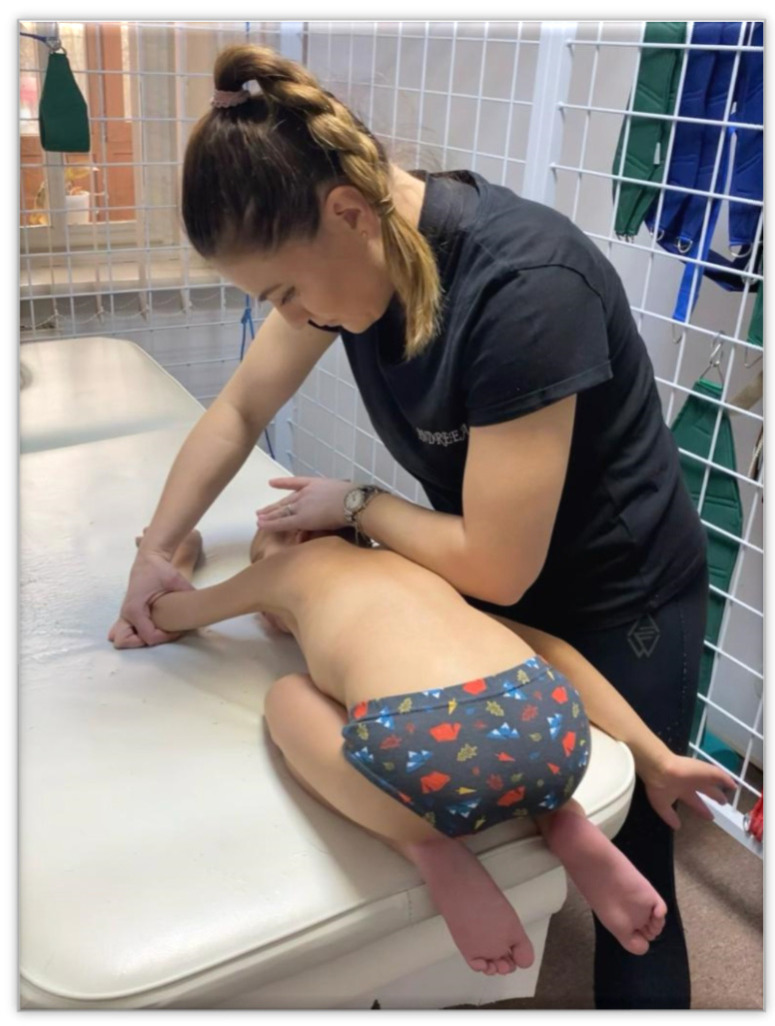
“First position”.

**Figure 5 children-09-01481-f005:**
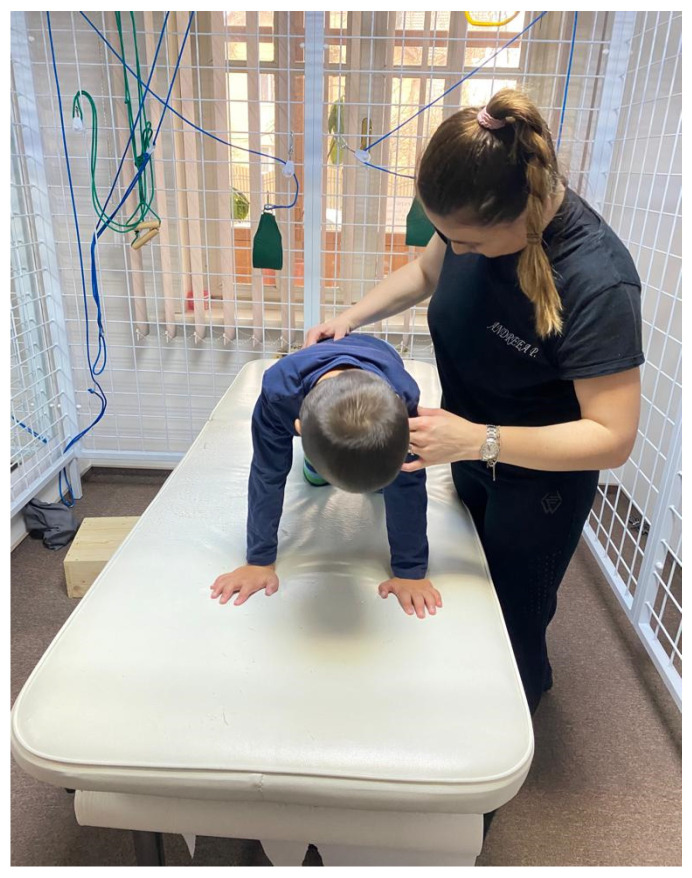
The exercise “Quadruped imbalances”.

**Figure 6 children-09-01481-f006:**
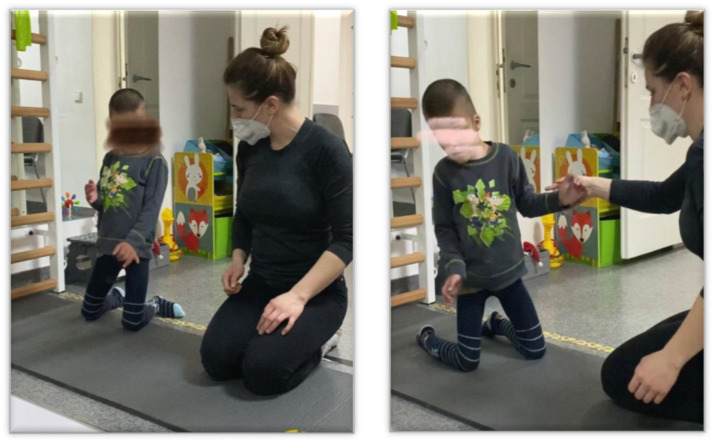
Imbalances from the “kneeling” position.

**Figure 7 children-09-01481-f007:**
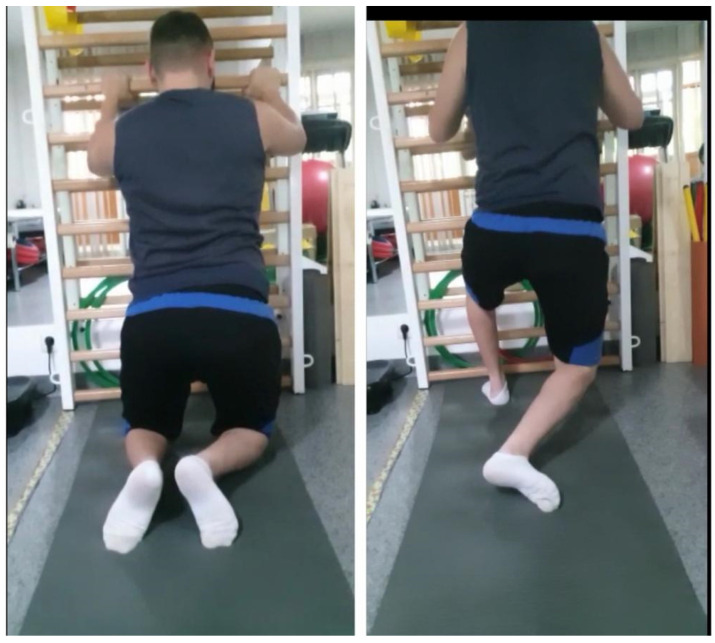
“The Servant Knight” position.

**Figure 8 children-09-01481-f008:**
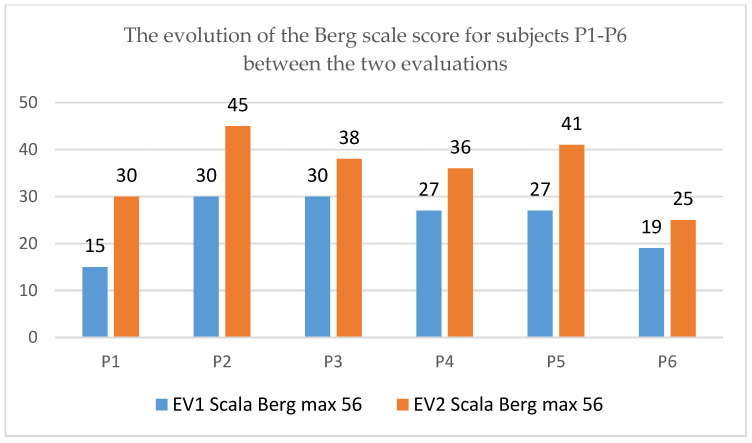
The evolution of the Berg scale score for subjects P1–P6.

**Figure 9 children-09-01481-f009:**
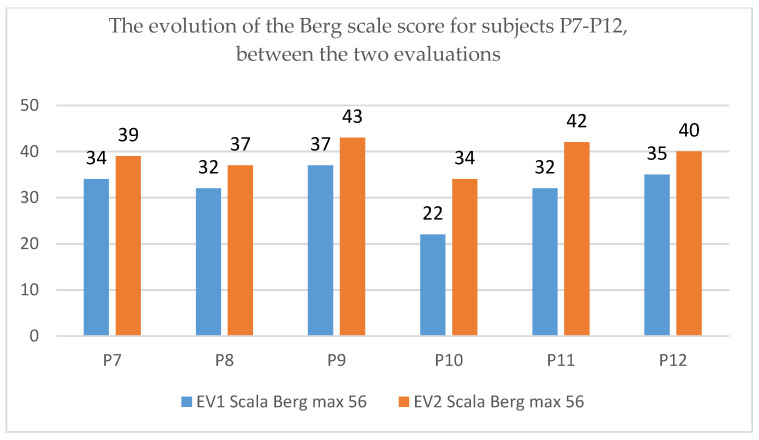
The evolution of the Berg scale score for subjects P7–P12.

**Figure 10 children-09-01481-f010:**
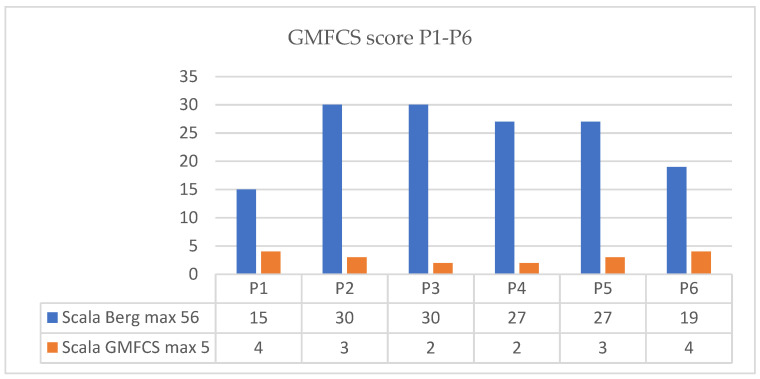
Evolution of GMFCS score patient P1–P6.

**Figure 11 children-09-01481-f011:**
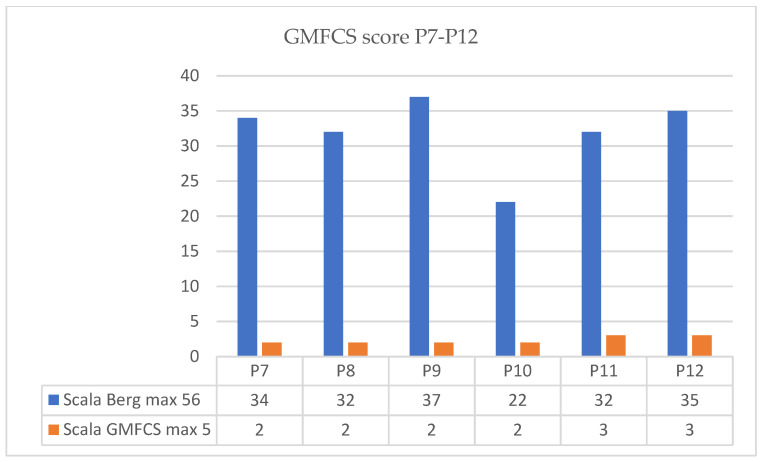
Evolution of GMFCS score patient P7–P12.

**Figure 12 children-09-01481-f012:**
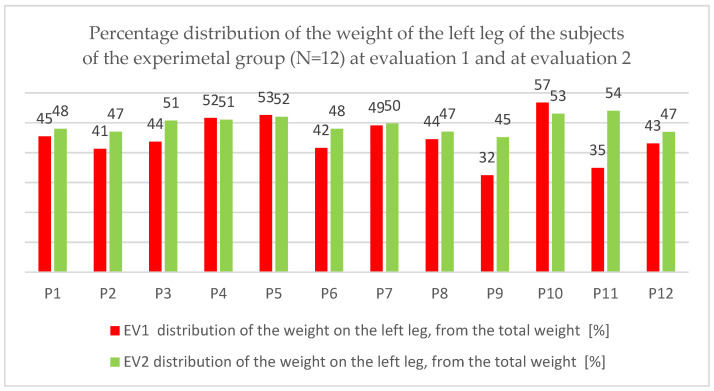
Percentage distribution of the weight on the left leg, from the total weight at the first and second evaluation of the subjects of the experimental group (N = 12).

**Figure 13 children-09-01481-f013:**
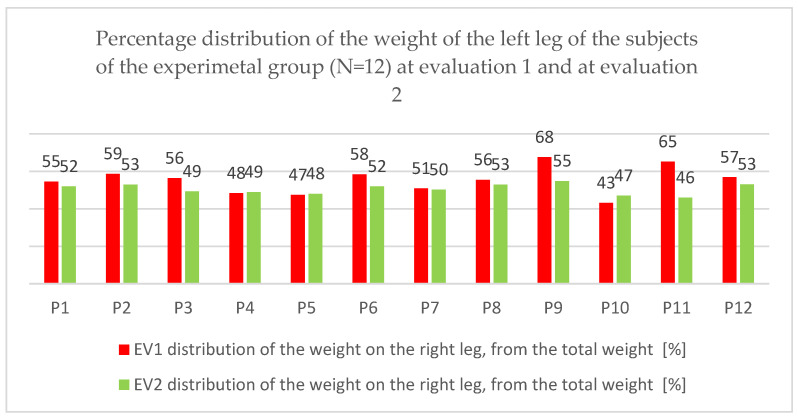
Percentage distribution of the weight on the right leg, from the total weight at the first and second evaluation of the subjects of the experimental group (N = 12).

**Table 1 children-09-01481-t001:** The values of the main anthropometric parameters of the subjects in the experimental group.

Subject Code (n = 12)	Age [Years]	Weight G [kg]	Height I [cm]	BMI = G/I*I [Kg/m^2^]
P1	7	28.20	121.00	19.31
P2	3	12.63	96.00	13.71
P3	10	44.50	131.00	25.93
P4	3	14.07	85.33	19.32
P5	6	17.43	103.75	16.19
P6	3	23.10	105.00	20.95
P7	11	56.23	151.67	24.45
P8	11	60.27	158.33	24.04
P9	10	27.90	136.00	15.08
P10	6	23.93	127.00	14.84
P11	4	20.70	108.33	17.64
P12	10	43.20	137.00	23.02

**Table 2 children-09-01481-t002:** The mean value of the Berg scale.

Test	Average	SD	Minimum	Maximum	CV
EV1	28.33	6.69	15	37	23.62
EV2	37.5	5.68	25	45	15.15

**Table 3 children-09-01481-t003:** The table of statistical indicators of the difference of the Berg scale.

Statistical Indicators of the Resulting Difference (EV2-EV1)			Bilateral Dependent Student *t*-Test			
Average	Standard Deviation	95% Confidence Interval	Effect Size (Cohen D’test)	t Obs	df	*p* *
9.16	3.97	(6.64; 11.69)	1.47	7.99	11	0.0001

* The significance threshold is *p* = 0.05.

**Table 4 children-09-01481-t004:** Table of average differences of the Berg scale.

Average Difference (EV1-EV2)	Progress	Difference Size	The Progress Is	Null Hypotesis (Averages Are Equal)
9.16	32.35%	Very high	Statistically significant	It is rejected

**Table 5 children-09-01481-t005:** Weight values and their distributions on each leg, at the two evaluations.

Patient Code	Total Weight (Kg)		Weight Distribution on Right Leg of Total Weight [%]		Weight Distribution on Right Leg of Total Weight [Kg]		Weight Distribution on Left Leg of Total Weight [%]		Weight Distribution on Left Leg of Total Weight [Kg]	
	EV1	EV2	EV1	EV2	EV1	EV2	EV1	EV2	EV1	EV2
**P1**	28.44	28.36	45	48	12.92	13.61	55	52	15.52	14.75
**P2**	11.72	12.93	41	47	4.84	6.08	59	53	6.88	6.85
**P3**	42.78	45.81	44	51	18.67	23.23	56	49	24.12	22.58
**P4**	12.76	15.33	52	51	6.59	7.82	48	49	6.18	7.51
**P5**	15.77	17.46	53	52	8.29	9.08	47	48	7.48	8.38
**P6**	19.37	23.65	42	48	8.05	11.35	58	52	11.31	12.30
**P7**	51.51	58.83	49	50	25.30	29.26	51	50	26.22	29.57
**P8**	57.35	64.37	44	47	25.50	30.26	56	53	31.84	34.12
**P9**	25.07	30.68	32	45	8.13	13.85	68	55	16.94	16.83
**P10**	22.67	24.07	57	53	12.87	12.75	43	47	9.80	11.31
**P11**	13.98	21.04	35	54	4.88	11.35	65	46	9.10	9.68
**P12**	42.74	43.80	43	47	18.41	20.54	57	53	24.32	23.26
**Min**	11.72	12.93	32.42	45.15	4.84	6.08	43.24	46.03	6.18	6.85
**Max**	57.35	64.37	56.76	53.97	25.50	30.26	67.58	54.85	31.84	34.12
**AV ***	28.68	32.19	44.74	49.37	12.87	15.77	55.26	50.63	15.81	16.43
**SD ****	15.95	17.11	7.06	2.78	7.46	8.14	7.06	2.78	8.80	9.04
**CV *****	55.62	53.13	15.78	5.62	58.00	51.62	12.77	5.48	55.64	55.00
**JB ******	0.521	0.504	0.908	0.685	0.532	0.502	0.908	0.685	0.553	0.510

* Average value. ** Standard deviation. *** CV is the coefficient of variation. **** Jarque-Bera test, calculated at the significance threshold of *p* = 0.05.

**Table 6 children-09-01481-t006:** The average value of the percentage distribution of the weight on the left leg.

Test	Average	SD	Minimum	Maximum	CV
EV1	55.26	7.06	43.24	67.58	12.77
EV2	50.63	2.78	46.03	54.85	5.48

**Table 7 children-09-01481-t007:** The table of statistical indicators of the difference in percentage distribution of weight on the left leg.

Statistical Indicators of the Resulting Difference (EV2-EV1)			Bilateral Dependent Student *t*-Test			
Average	Standard Deviation	95% Confidence Interval	Effect Size (Cohen D’test)	t obs	df	*p* *
4.64	6.29	(0.63; 8.63)	0.86	2.552	11	0.027

* The significance threshold is *p* = 0.05.

**Table 8 children-09-01481-t008:** Table of average differences of percentage weight distribution on the left leg.

Average Difference (EV1-EV2)	Progress	Difference Size	The Progress Is	Null Hypotesis (Averages Are Equal)
4.64	8.39%	Average	Statistically significant	It is rejected

**Table 9 children-09-01481-t009:** The average value of the percentage distribution of the weight on the right leg.

Test	Average	SD	Minimum	Maximum	CV
EV1	44.74	7.06	32.42	56.76	15.78
EV2	49.37	2.78	45.15	53.97	5.62

**Table 10 children-09-01481-t010:** The table of statistical indicators of the difference in percentage distribution of weight on the right leg.

Statistical Indicators of the Resulting Difference (EV2-EV1)			Bilateral Dependent Student *t*-Test			
Average	Standard Deviation	95% Confidence Interval	Effect Size (Cohen D’test)	t Obs	df	*p* *
−4.64	6.29	(8.63; 6.37)	0.86	2.552	11	0.027

* The significance threshold is *p* = 0.05.

**Table 11 children-09-01481-t011:** Table of average differences of percentage weight distribution on the right leg.

Average Difference (EV1-EV2)	Progress	Difference Size	The Progress Is	Null Hypotesis (Averages Are Equal)
**−4.64**	**10.36%**	**Big**	**Statistically significant**	**It is rejected**

## Data Availability

Not applicable.

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
