# Peer review of "Balance Rehabilitation Approach by Bobath and Vojta Methods in Cerebral Palsy: A Pilot Study"

_children, 2022, doi:10.3390/children9101481_

Round 1

Reviewer 1 Report

The manuscript topic is of interest and clinical relevance. Overall, the manuscript content is good, provided sufficient introduction and research design. At present, some minor concerns related to grammar, clarity of methods section, and limitations of the current studies need to be improved.

Title: Balance rehabilitation approach by Bobath and Vojta methods 2 in cerebral plasy

Reviewer comments:

Page 1 line 42-44: Provide/add reference for this statement “Neurons have the ability to change their structure and function depending on the inputs generated by activity and learning; in fact, neural change is the basis for memory and behavioral change that results from experience”.

Page 1 line 44: Provide/add reference for these two statements “Plasticity takes place constantly, regardless of whether we are subjected to intense training or not. In addition, plasticity can be positive (adaptive) or negative (maladaptive). The central nervous system (CNS) has an innovative capacity to recover and to adapt to the compensatory mechanisms following an injury”.

Page 2 line 48: Provide/add reference for this statement “The basis for rehabilitation comes from neuroplasticity, defined as the ability of neural network to make adaptive changes at both structural and functional levels, ranging from molecular, synaptic and cellular 50 changes to global network changes”.

Page 2 line 53: Add “ and they can recover function after injury/insult to any of these structures by applying”.

Page 2 line 57: change [3], [4] to [3,4].

Page 2 line 58: change “ can” to “may involve an”.

Page 2 line : Please provide/add reference for these statements “Cerebral palsy (CP) is a pathology that can involve an interruption of the descending tracts. It is caused by a non-progressive disorder of the brain that occurred during the fetal or developing infant period. This disorder can cause certain permanent limitations of posture and movement”

Page 2 line 63: Merge this paragraph with previous paragraphs (not as a separate paragraph/no gap).

Page 2 line 64: Replace “physical therapy” with “rehabilitation”

Page 2 line 71-72: Provide/add reference for this statement on Bobath and Vojta therapy.

Page 2 line 84: Remove the word “very”.

Page 2 line 91: Provide/add reference for this statement “Bobath therapy (Neurodevelopmental Therapy), which is based on the fact that the atypical development concerning postural control and reflexes is responsible for the observed motor anomalies, due to the basic dysfunction of the central nervous system”.

Page 2 line 98 and page 3 line 107: Please be consistent with using abbreviations. For example, CNS and NDT, and CP (page 112) words are sometimes abbreviated.

Page 3 line 108: Please rewrite this statement to reflect accurate findings from Grazziotin Dos Santos (2015). This work concluded that the NDT (one specific technique) increased activation of trunk extensor muscles to facilitate trunk stability. You could rewrite this statement as” Despite its widespread use in the clinic, limited studies have demonstrated evidence for its effectiveness in children with CP. Thus, a valid and reliable assessment tools are needed to measure the effectiveness of therapy in CP rehabilitation

Page 3 line 125-138: Please remove this paragraph (Hyungwon Lim proposed) as this does not add to your study question or move it up few lines above to embed it with other studies on NDT showing limited evidence of its effectiveness. If you decide to add/embed this study, please provide a context rather than just explaining aim and method. For example, please add what were the findings and limitations of this study (small sample size) and how additional studies with bigger sample size and more controlled interventions are needed to study effectiveness of Vojta and Bobath therapies.

Page 3 line 145: Add the word “study” after “our research”

Page 4 line 200: Was medication (for spasticity) in exclusion criteria or if any of the participants were on medication that could influences muscle tone etc. Any surgery (tendon transfer etc.).

Page 5 line 207: Provide reference for this statement “ The clinical functional evaluation of the 12 subjects included the balance assessment using the Berg scale” (Reference of this scale used in literature)

Page 5 line 208-216: Provide reference on use of this device in children (if used in studies in children or adults to measure body weight distribution). 

Page 5 line 218: Re-write the sentence to “At the time of evaluation, we accounted for the fact that the CNS…”

Page 6 line 258 Re-write this sentence to “For the Bobath technique/method, exercises were performed in supine position (SP), lateral position (LP), prone position (PP), on all fours, on both knees, sitting on the edge of the bed or sitting on the chair, the half kneeling position, and orthostatism position. For the Vojta method, exercises that require activation positions in SP, LP, PP, and the first position (crouching at the edge of the bed) were performed.

Page 4 line 259: Use past tense to describe methodology. This study already completed so, avoid describing it in continuous present tense.

Page 5 line 205: Please expand term PCI

Page 5 line 269: Re-write the sentence to “ On first session, patient assessment was performed using scale application and PoData platform.

Page 5 line 273 and 276: Use past tense for rest of methodology “ The first session also aimed to initiate the exercise program” The physiotherapist provided information”.

Page 5 line 224: Alone or assisted: please clarify if data was collected with them on platform assisted or participants were only assistant to get to the platform and data was recorded with every participant standing unassisted? If not that could influence the weight distribution reading based on where and how much assistance was provided.

Page 16 line 535-537: Re-write this line as “A study by Moazma Jamil compared effects of convention and Bobath therapy to improve GMF among 24 children with CP. Children with CP received 3 months of intervention, 40minutes/day, 5days/week”. All children were tested…

Page 16 line 539: This sentence is too long. Break it into two or three sentence.

Page 16 line 544: Discuss these points under discussion and or under limitations section.

 Please address these points in your discussion/limitations.

1.      What about severity of CP participants in this study? How many of them had a score of I, II, III, or IV GMFCS.

2.      What about differences in degree of improvements in each participant.

3.      Did this study account for participants on medication for spasticity? If yes or no, please explain. In addition, the current study did not measure spasticity in participants. That could have touched on potential mechanisms of this improvement.

4.      How many participants needed assistance during loading on platform? Were they standing independently, or support was provided during measurement of weight distribution? If so, could that influence the results?

5.      Please add a few sentences as to what is the clinical significance of these findings? Should therapists use a combination of these two techniques? Can this be implemented to children with severed CP? Please expand on this a bit as to what information you want your fellow Physical therapists to know based on your findings. Will this help them designing their intervention?

Page 17 line 579: Reference format is not accurate. All author names should be listed (as it is done for other references). Same for reference number 4, 7, 8, and 11. Please fix this.

Thank you,

Reviewer

Author Response

Thank you very much for your useful comments.We try to answer and to do the best for improve our paper.

Reviewer 2 Report

Dear Authors,

I am delighted to read your manuscript and see the results of your research. The topic is very interesting and valuable. I also appreciated the long duration of the therapy and the work you had to do.

In my opinion, the weak point of the research is the lack of a control group. I don't think about the group of children with CP who are not undergoing rehabilitation, because that shouldn't happen. It was ideal to see if Bobath therapy alone or Vojta therapy alone would have similar effects to a combination of the two. This would be of high practical value. Parents of children with CP would know if it is worth looking for a therapist who can perform both methods, and a therapist working with one of these methods would know if it is worth spending time and money to learn the other method as well. Or would he rather improve in the method he or she is already working with patients?

Below are some minor remarks.

Best regards and good luck in your further research work.

Lines 154-156: We studied a number of 12 subjects diagnosed with CP, aged between 3-11yrs. The small number of subjects is due to the fact that the consistent participation of these children in a rehabilitation program is limited. Therefore, we cannot cover a complete application of rehabilitation program.

Could you write, please, how the recruitment of patients to the project was carried out, how many children started the project, how many dropped out during and what were the reasons for resignation?

Lines 157-158: The group of subjects included 4 boys and 8 girls, with an average age of 7±3.28 years, average height of 121.7±22.48 cm, average weight of 28.68±15.95 Kg (mean value±SD).

I am not sure if there is a point in giving the average body height and average body weight if the patients differed so much in age? Does this information add anything? The data in Table 1 seems to be sufficient.

Lines 162-163: In the same time, we have the informed consent for all children that have been included in the research.

From 3-year-olds too? Is this not an overinterpretation? Perhaps a statement that parents have given written consent for their children to participate in the study is sufficient?

Line 193: children aged between 3-11 years;

Was it really a condition, or did it just happen? It wouldn't be a bad thing, after all. Would a 12-year-old child who met all other criteria be excluded from the study?

Line 224: The patient must be without shoes, the skin must be bare.

I think in Figure 2 the patient has a Kinesio tape application.  Shouldn't this be treated as additional therapy and noted?

Author Response

Thank you very much for your kindly words and feedback.

We try to do the best in our paper, but how you know is very difficult to work and to make research with children suffer of CP. We have not in this moment a control group because is very hard to have a permanently group. Of course you have right could be better to extent our research and make the comparison between single therapy0Vojta, Bobath and combination, but again we have possibility to be sure that we have a permanently participation. We can consider that our research is a pilot study or preliminary research, and we extent the research.

We try to answer to your questions and hope that our answers will response to your comments. Also hope that you will give us a positive feedback.

Thank you very much for time spent for evaluate our paper.

Reviewer 3 Report

The authors have undertaken to present an interesting topic, but the paper needs to be revised. I would ask the authors to address these.

Author Response

Thank you for time spent for analysis our paper and give very good suggestion for improve the quality of our research.

We try to do the best and give the answers for your comments. Also we think that we shall have a positive feedback from you.

Thank you again.

Round 2

Reviewer 2 Report

Thank you for the changes made. It would be good to make it clear that this is a pilot study.

Author Response

Thank you very much for your comments that  contribute to improve the paper quality.

Reviewer 3 Report

The authors addressed most of the comments.

Author Response

Thank you again for your effort in assessment our paper and also for your comments.
